# Discovery of Novel Allosteric Modulators Targeting an Extra-Helical Binding Site of GLP-1R Using Structure- and Ligand-Based Virtual Screening

**DOI:** 10.3390/biom11070929

**Published:** 2021-06-23

**Authors:** Qingtong Zhou, Wanjing Guo, Antao Dai, Xiaoqing Cai, Márton Vass, Chris de Graaf, Wenqing Shui, Suwen Zhao, Dehua Yang, Ming-Wei Wang

**Affiliations:** 1School of Basic Medical Sciences, Fudan University, Shanghai 200032, China; zhouqt@fudan.edu.cn; 2The National Center for Drug Screening, Shanghai Institute of Materia Medica, Chinese Academy of Sciences, Shanghai 201203, China; 15250959828@163.com (W.G.); atdai@simm.ac.cn (A.D.); xqcai@simm.ac.cn (X.C.); 3University of Chinese Academy of Sciences, Beijing 100049, China; 4The CAS Key Laboratory of Receptor Research, Shanghai Institute of Materia Medica, Chinese Academy of Sciences, Shanghai 201203, China; 5Amsterdam Institute for Molecules, Medicines and Systems, Division of Medicinal Chemistry, Faculty of Sciences, Vrije Universiteit Amsterdam, 1081 Amsterdam, The Netherlands; marciavassklanbol@gmail.com (M.V.); Chris.DeGraaf@SoseiHeptares.com (C.d.G.); 6iHuman Institute, ShanghaiTech University, Shanghai 201210, China; shuiwq@shanghaitech.edu.cn; 7School of Life Science and Technology, Shanghai Tech University, Shanghai 201210, China

**Keywords:** GLP-1R, virtual screening, allosteric modulator, drug discovery, molecular docking

## Abstract

Allosteric modulators have emerged with many potential pharmacological advantages as they do not compete the binding of agonist or antagonist to the orthosteric sites but ultimately affect downstream signaling. To identify allosteric modulators targeting an extra-helical binding site of the glucagon-like peptide-1 receptor (GLP-1R) within the membrane environment, the following two computational approaches were applied: structure-based virtual screening with consideration of lipid contacts and ligand-based virtual screening with the maintenance of specific allosteric pocket residue interactions. Verified by radiolabeled ligand binding and cAMP accumulation experiments, two negative allosteric modulators and seven positive allosteric modulators were discovered using structure-based and ligand-based virtual screening methods, respectively. The computational approach presented here could possibly be used to discover allosteric modulators of other G protein-coupled receptors.

## 1. Introduction

G protein-coupled receptors (GPCRs) influence virtually every aspect of human physiology [1,2] and are one of the most successful therapeutic targets with over 500 approved drugs [3,4]. To mediate transmembrane signal transduction, GPCR has the following two spatially distant but conformationally linked regions: the extracellular agonist-binding site and the intracellular transducer binding site [5,6]. The agonist binding in the extracellular side promotes the transmembrane domain (TMD) of GPCR to undergo extensive conformational changes and eventually activates the intracellular transducers, such as G protein and β-arrestin. Besides the orthosteric sites where the endogenous agonists bind, recent structural and pharmacological studies highlight the fact that ligands can bind spatially and topologically to distinct (allosteric) sites on receptors and modulate GPCR-mediated signaling simultaneously through conformational cooperativity [6,7,8,9,10,11,12]. Allosteric modulators that enhance the functional response of an orthosteric agonist are regarded as positive allosteric modulators (PAMs), while those that inhibit or negatively modulate the action of an orthosteric ligand are called negative allosteric modulators (NAMs) [13].

The glucagon-like peptide-1 (GLP-1) receptor (GLP-1R) belongs to the class B1 GPCR subfamily and plays a crucial role in glucose homeostasis. As a successful therapeutic target for type 2 diabetes and obesity, many peptidic analogs of GLP-1 are on the market. Meanwhile, continuous efforts in small-molecule drug discovery resulted in several non-peptidic agonists such as Boc5 [14], TT-OAD2 [15], PF-06882961 [16], RGT1383 [17] and HD-7671 [18], as well as dozens of small molecule allosteric modulators that interact with distinct regions of GLP-1R. There are at least four reported allosteric sites in class B1 GPCRs including: (i) deep inside the helical bundle observed in the corticotropin-releasing factor receptor type 1 receptor (CRF1R) [19]; (ii) extracellular helical bundle at the TM helices 1 and 2 interface found in GLP-1R [8]; (iii) TM helices 3−4−5 interface with receptor activity-modifying protein 1 (RAMP1) seen in the calcitonin gene related peptide receptor (CGRP) [20]; and (iv) outside of the helical bundle of TM helices 5, 6 and 7 at the lipid interface of GLP-1R and glucagon receptor (GCGR) [7,21]. Interestingly, two PAM agonists of GLP-1R, 6,7-dichloro-3-methanesulfonyl-2-tert-butylamino-quinoxaline (compound 2) and 4-(3-benzyloxyphenyl)-2-(ethylsulfinyl)-6-(trifluoromethyl) pyrimidine (BETP) [6,7,22], have been found to covalently link with C347^6x36b^ (Wootten numbering in superscript) [23] at the intracellular end of TM6, sharing a similar region as the NAMs (such as NNC0640 and PF-0637222) [7,21]. In the case of GLP-1R, a variety of allosteric modulators have been described [24,25], including compound 19 [26], ZINC19797057 [27], compound 3286 [28], CD3878-F005 [29], HTL26119 [30] and ZINC00702587 [31], thereby greatly enhancing our knowledge about the small molecule allosterism of this important drug target.

Assisted by advances in algorithm, protocol and software development, computer-aided drug design (CADD) has shown promise in the discovery of novel drug candidates targeting GPCRs [32,33]. Structure-based virtual screening (SBVS) and ligand-based virtual screening (LBVS) are two distinct CADD approaches. SBVS employs three-dimensional (3D) structural information of a target protein and performs molecular docking to identify potent binders, while LBVS utilizes known active ligands to establish a structure-activity relationship (SAR) for guiding subsequent lead discovery and optimization. Given that the scoring function in SBVS is developed for soluble proteins without considering the membrane environment, and the number of known allosteric modulators for LBVS is limited, virtual screening that aims to discover allosteric modulators for GPCRs is still challenging.

Here, we report the discovery of allosteric modulators of GLP-1R by computational screening and experimental validation. Focused on an extra-helical allosteric site (outside of the intracellular half of TMs 5-6-7 at the lipid interface) where the NAM PF-0637222 and PAM compound 2 bind, we developed a new virtual screening strategy to discover allosteric modulators at the lipid interface of GPCR TMD: SBVS takes the lipid interactions into consideration and LBVS retains the specific residue contacts involved in the allosteric site. Verified using cAMP accumulation and radiolabeled ligand binding assays, two NAMs and seven PAMs were discovered using these two methods, respectively.

## 2. Materials and Methods

### 2.1. Ligand Database

Chemical structures from Enamine, ChemDiv, Vitas-M, ChemBridge and TimTec (~540,000 compounds) as well as the Chinese National Compound Library (CNCL, ~760,000 compounds) were collected for virtual screening. The reported NAMs (PF-06372222, NNC0640 and MK0893) that interact with the extra-helical binding sites of GLP-1R or GCGR share similar chemotypes and one anionic end (carboxylic acid or tetrazole) inserts into a polar cleft between TM6 and TM7. After being protonated at a pH of 7.4 and its properties calculated using ChemAxon (cxcalc 5.1.4, Budapest, Hungary), the compound collection was filtered with the following criteria: (i) 18 ≤ HAC (number of heavy atoms) ≤ 36; (ii) 1 ≤ HBA (number of hydrogen bond acceptors) ≤ 10; (iii) 0 ≤ HBD (number of hydrogen bond donors) ≤ 5; (iv) 0 ≤ clogP ≤ 5; and (v) no positively charged atoms and at least one negatively charged atom or containing an acid isostere [34]. A substructure search was then performed using the ChemAxon (cxcalc 5.1.4), resulting in 27,939 compounds having negatively charged atoms, 16,103 compounds with isostere from CNCL, as well as 195,831 and 87,873 compounds from commercially available libraries, respectively. In addition, 583 CNCL compounds and 11,779 commercially available compounds were identified as they have alternative chirality. Three-dimensional conformations of these compounds were generated using Corina (4.1.0, Erlangen, Germany) [35] with default parameters, except for the maximum number of generated stereoisomers per molecule, which was set to 1 to restrict the number of output stereoisomers. Finally, a tailored library consisting of 340,108 compounds was prepared for virtual screening.

### 2.2. Protein Preparation

The crystal structure of human GLP-1R TMD in a complex with two negative allosteric modulators [7], PF-06372222 (PDB code: 5VEW) and NNC0640 (PDB code: 5VEX), were obtained from the Protein Data Bank [36]. The missing side chains and hydrogens were added and optimized using Protein Preparation Wizard (New York, NY, USA) [37] in Schrödinger Suite 2017-3. Given the lipid environment for the extra-helical binding site of GLP-1R, we embedded the receptor within the palmitoyl oleoyl phosphatidyl choline (POPC) bilayer using PyMOL (v1.7, New York, NY, USA) [38,39] and performed short-time molecular dynamics (MD) simulation to relax the protein and lipid molecules. The force field parameters of two ligands (PF-06372222 and NNC0640) were modelled with ACPYPE (Wilmington, DE, USA) [40], while the CHARMM36-CAMP force field [41] was applied to the receptor and lipids. MD simulations were conducted using Gromacs (5.0.2, Groningen, The Netherlands) [42]. All bonds involving hydrogen atoms were constrained using LINCS algorithm [43]. The particle mesh Ewald (PME) method was used to treat long-range electrostatic interactions with a cutoff of 10 Å. The entire system was first relaxed using the steepest descent energy minimization, followed by equilibration steps of 5 ns in total to equilibrate the lipid bilayer and the solvent, while the position restraints to the protein and the ligand were retained. As we focused on the extra-helical binding site of GLP-1R, these POPC molecules within 5 Å of the ligand were kept, while the rest of lipids, water and ions were removed.

### 2.3. Structure-Based Virtual Screening

Receptor grids were generated for the two complexes using the Receptor Grid Generation tool (New York, NY, USA) in the Glide module [44] of Schrödinger Suite 2017-3. The grid boxes were defined as a 10 × 10 × 10 Å^3^ region centered at negative allosteric modulators PF-06372222 or NNC0640. For virtual screening, the prepared 340,108 compounds were subjected to the following two levels of docking using the relevant workflow in Glide [44]. After a standard precision (SP) docking stage, compounds with top 10% docking scores went through an extra precision (XP) docking process by adopting a more accurate scoring function essential to reduce false positives. To further relax the complexes and approximately calculate the binding free energies, the XP docking poses with the top 1,000 docking scores were subjected to a Prime MM-GBSA calculation [45]. All residues within 7 Å of the docked ligands were relaxed with the sampling method “Minimize”. Finally, by visual inspection of the optimized docking pose and considering the XP docking scores, calculated binding free energies (MM-GBSA dG Bind) and chemotype diversity, 45 compounds were selected and purchased from TargetMol (Boston, MA, USA) with a specified purity of >95% for experimental validation (Appendix A).

### 2.4. Ligand-Based Virtual Screening

Ligand-based dockings were performed using Protein-Ligand ANT System (PLANTS) [46] version 1.2 (Konstanz, Germany), post-processed and ranked by Interaction Fingerprints (IFP) [47]. PLANTS employs an ant-colony-optimization algorithm for the prediction of binding poses of small molecules in a protein structure and an empirical scoring function, ChemPLP, for grading the resultant binding poses. The docking site of GLP-1R was defined by all residues within a 5 Å radius around the co-crystallized ligands (PF-06372222 and NNC0640) [7], and for each ligand, 25 poses were produced (speed setting 2) and scored using the ChemPLP scoring function [46]. IFP evaluates a (predicted) binding mode of a compound in a protein structure by annotating the presence or absence of different types of interactions (hydrophobic, aromatic, H-bond, ionic) between each pocket residue and the molecule based on a set of rules [47]. This results in a molecular interaction fingerprint representing all interactions between the molecule and the protein in bit-string, allowing for easy comparison and scoring (using the Tanimoto coefficient) of the similarity of multiple IFPs. In our previous virtual screening toward orthosteric sites of the histamine H_1_ receptor (H_1_R) [48], the IFP score (≥0.75) and PLANTS score (≤−90) cutoffs are able to discriminate H_1_R ligands from decoys with the discovery of a chemically diverse set of novel fragment-like H_1_R ligands. A similar screening strategy was applied to β2-adrenoceptor with a hit rate of 53% and hits with up to nanomolar affinities and potencies [49]. Considering that the allosteric modulator-binding site in the present study is on the lipid facing surface where the ligands have more freedom in the docking simulation, a lower PLANTS score (≤−75) and IFP similarity (IFP score of ≥ 0.6) cutoff were applied to obtain a reasonable number of top hits from the virtual screening. The allosteric GLP-1R pocket was defined by 41 pocket residues within 6.5 Å around the co-crystallized ligands [7] (R^2x46b^, H^2x50b^, L^3x54b^, F^5x54b^, V^5x57b^, I^5x58b^, V^5x61b^, V^5x62b^, L^5x65b^, M340, K342, D^6x33b^, I^6x34b^, K^6x35b^, F^6x36b^, R^6x37b^, L^6x38b^, A^6x39b^, K^6x40b^, S^6x41b^, T^6x42b^, L^6x43b^, T^6x44b^, L^6x45b^, I^6x46b^, P^6x47b^, L^6x48b^, F^7x44b^, L^7x51b^, M^7x52b^, I^7x55b^, L^7x56b^, Y^7x57b^, C^7x58b^, F^7x59b^, V^7x60b^, N^8x47b^, N^8x48b^, E^8x49b^, V^8x50b^ and Q^8x51b^). Ionic interaction distance cutoff was set to 6.5 Å. In the GLP-1R retrospective validation, the binding mode of the co-crystallized compound for each respective crystal structure [7] was used for calculation of a reference IFP. These reference IFPs were subsequently used to score the docking poses. Filtering was performed by applying two filters (polar interactions with S^6x41b^ and N^8x47b^, and contacts with at least one of V^5x61b^, F^6x36b^ and L^6x43b^), the PLANTS (PLANTS score of ≤−75 according to the best IFP pose) and IFP cutoff (only compounds with an IFP score of ≥0.6 according to the best PLANTS pose). Compounds were visually clustered based on scaffold similarity to ensure structural diversity, and those with buried polar groups located in hydrophobic parts of the receptor binding site were discarded by visual inspection. Finally, 42 compounds, including one compound shared by LBDD (ZINC72191544), were chosen and purchased from TargetMol (Boston, MA, USA) with a specified purity of >95% for experimental validation (Appendix A).

### 2.5. cAMP Accumulation Assay

cAMP accumulation was measured using the LANCE Ultra cAMP kit (PerkinElmer, Boston, MA, USA) according to the manufacturer’s instructions. CHO-K1 cells stably expressing wild-type (WT) or mutant GLP-1R were digested by 0.2% (*w/v*) EDTA and washed once with PBS. Cells were then resuspended with assay buffer (DMEM, 0.1% BSA, 5 mM HEPES) and seeded onto 384-well plates (6 × 10^5^/mL, 5 μL/well). Transfected cells were incubated for 40 min with 20 μM allosteric modulators and different concentrations of GLP-1. After the addition of 5 μL of Eu-cAMP tracer and ULight-anti-cAMP, the reactions were stopped. The plates were left for 1 h at room temperature to measure time-resolved FRET signals at excitation wavelengths of 620 nm and 665 nm by EnVision (PerkinElmer, Waltham, MA, USA). The cAMP response is depicted relative to the maximal response of GLP-1 at the WT receptor (100%).

### 2.6. Whole Cell Binding Assay

CHO-K1 cells stably expressing WT or mutant GLP-1R were seeded into 96-well plates at a density of 3 × 10^4^ cells/well and incubated overnight at 37 °C and 5% CO_2_. For homogeneous binding, cells were washed twice and incubated with a blocking buffer (F12 supplemented with 33 mM HEPES, and 0.1% (*w/v*) BSA, pH 7.4) for 2 h at 37 °C. Then, radiolabeled ^125^I-GLP-1 (40 pM, PerkinElmer, Waltham, MA, USA) and unlabeled compounds were added and reacted with the cells in binding buffer (F12 supplemented with 33 mM HEPES, and 0.1% (*w/v*) BSA, pH 7.4) at 4 °C overnight. Cells were washed three times with ice-cold PBS and lysed using 50 μL of lysis buffer (PBS supplemented with 20 mM Tris–HCl and 1% Triton X-100, pH 7.4). Subsequently, the plates were counted for radioactivity (counts per minute, CPM) in a scintillation counter (MicroBeta^2^ Plate Counter, PerkinElmer, Waltham, MA, USA) using a scintillation cocktail (OptiPhaseSuperMix; PerkinElmer, Waltham, MA, USA).

## 3. Results

### 3.1. Virtual Screening Workflow

A schematic workflow of the virtual screening toward the allosteric modulator for GLP-1R is presented in Figure 1. The high-resolution X-ray structures of the human GLP-1R in complex with negative allosteric modulators PF-06372222 and NNC0640 were adopted [7,37]. After analyzing the reported allosteric modulators, we screened a focused library with loosely defined physicochemical properties that were filtered as described above [34]. SBVS and LBVS were performed independently with an identical input (chemical library and receptor file). For SBVS, three stages were adopted. In the first stage, the compounds were docked using Glide (New York, NY, USA) standard precision (SP) [44] where the top 10% of good scoring compounds were retained. Then, these compounds were docked with Glide extra precision (XP) where the top 1000 hits of the best docking scores were kept. Finally, these hits were processed by Prime MM-GBSA [45] to further relax the complex and calculate the binding free energy. For LBVS, three filtering protocols were applied. Firstly, these compounds were docked to the receptor with PLANTS [50,51], which predicts and scores the binding poses in a protein structure by a combination of an ant colony optimization algorithm with an empirical scoring function. The resultant docking poses of 10,000 compounds were subjected to molecular interaction fingerprint (IFP) calculations, where seven different interaction types (negatively charged, positively charged, H-bond acceptor, H-bond donor, aromatic face-to-edge, aromatic face-to-face and hydrophobic interactions) were adopted to define the IFP. The top 100 compounds with the best IFP scores were further screened with the pan-assay interference compound filter (PAINS) [52] to exclude reactive functional groups and promiscuous chemotypes. Based on our previous LBVS experience [48,49] and the consideration that the docked ligands in the allosteric modulator-binding site may have more freedom in the lipid environment, the PLANTS score (≤−75) and IFP similarity (IFP score of ≥0.6) cutoff were applied. To maximize the chemotype diversity, these compounds were classified into 42 clusters using spectral clustering, and the potency of the compounds in each cluster was further assessed by visual inspection of their interactions with the residues that constructed the allosteric extra-helical binding site of GLP-1R. Eventually, 45 compounds identified using SBVS and 42 compounds found using LBVS were purchased and underwent experimental validation (Appendix A).

### 3.2. MD Simulation Studies

Given that the accuracy of molecular docking is fundamental for high-quality virtual screening, it is important to redock NNC0640 and PF-06372222 to the allosteric site of GLP-1R. Two different ligand-sampling methods, score in place (not docking and use the original position for scoring) and flexible docking (generate conformations internally and then perform molecular docking and scoring), were applied during Glide XP docking, thereby reflecting the scoring accuracy and sampling efficiency, respectively. As shown in Figure 2, the Glide XP (New York, NY, USA) docking scores of the crystal pose of NNC0640 in GLP-1R [7] by score in place and flexible docking are −2.9 and −4.1 kcal/mol (a more negative docking score indicates stronger binding), respectively. A similar observation was made for PF-06372222, whose docking scores are −4.8 and −4.6 kcal/mol, respectively. Moreover, the root-mean square deviation (RMSD) of the redocked poses relative to the crystal pose [7] is 5.5 Å for NNC0640 and 5.2 Å for PF-06372222 (Figure 2A), suggesting that traditional molecular docking failed to reproduce the crystal poses [7]. In addition, Prime MM-GBSA calculation demonstrates that both ligands showed weak binding energies (MM-GBSA dG Bind, around −50 kcal/mol). As a comparison, the orthosteric ligands with nanomolar binding affinity in class A GPCRs generally have a docking score and MM-GBSA dG Bind lower than −10.0 and −100.0 kcal/mol, respectively.

To mimic the lipid environment of the extra-helical binding site of GLP-1R during molecular docking, we performed short-time MD simulations for NNC0640 or PF-06372222 bound GLP-1R placed in a POPC bilayer that was retained within 5 Å of the ligands (Figure 2A). With the assistance of POPC, the Glide XP docking scores of both were improved to −11.1 and −10.9 kcal/mol, respectively (Figure 2B), significantly better than that without lipids. In addition, their docked poses are almost identical to that observed in crystal structures [7] with RMSDs of 1.1 Å and 0.3 Å, respectively. A similar improvement was noted with the Prime MM-GBSA binding free energy, whose values were −84.9 and −86.3 kcal/mol for NNC0640 and PF-06372222, respectively (Figure 2B). Collectively, these results demonstrate the key role of lipids in generating correct docking conformations and reasonable docking scores.

### 3.3. Identification of New NAMs

In SBVS, 113 compounds have Glide XP (New York, NY, USA) docking scores that are better than −10.0 kcal/mol, and 11 of them are lower than −11.0 kcal/mol, comparable to that of NNC0640 (−11.1 kcal/mol) and PF-06372222 (−10.9 kcal/mol). However, none have a better binding free energy (MM-GBSA dG Bind) than that of NNC0640 (−84.9 kcal/mol) or PF-06372222 (−86.3 kcal/mol). Experimental validation found that two of them, Z21 (ZINC254697034) and Z42 (ZINC16949012), negatively modulated GLP-1-elicited cAMP accumulation (Figure 3 and Figure 4A, Table 1), where Z21 has an EC_50_ of 76 μM. In the binding assay, Z21 decreased the binding of GLP-1 to GLP-1R (Figure 4B).

Figure 3 shows that Z21 and Z42 were predicted to occupy the extra-helical binding site in a manner similar to NNC0640 and PF-06372222. These four compounds share a common negatively charged terminus that inserts into a positively charged cleft between TM6 and TM7 and forms multiple hydrogen bonds or a salt bridge with the polar residues (such as R176^2x46b^, N406^8x47b^ and N407^8x48b^). In terms of other segments, Z21 and Z42 make weaker interactions with GLP-1R compared to NNC0640 or PF-06372222. The residues in TM6 form at least two hydrogen bonds with both PF-06372222 and NNC0640 via S352^6x41b^ and T355^6x44b^, but only one hydrogen bond was observed for Z21 (via S352^6x41b^) or Z42 (via T355^6x44b^). Moreover, PF-06372222 and NNC0640 have extensive hydrophobic contacts with I328^5x58b^, V331^5x61b^, A350^6x39b^ and L354^6x43b^ through their extended arms pointing to TM5. Such contacts are much weaker for Z21 (short arm) and Z42 (without arm). The differences in the binding modes and efficiencies between potent NAMs (PF-06372222 and NNC0640) and weak NAMs (Z21 and Z42) suggest that the polar interaction contributed by TM6 and the hydrophobic contacts from TM5 are essential for negative allosteric modulation, an observation supported by our mutagenesis studies (Figure 4B), where C347F enhanced the ligand binding of Z42. These data are valuable to structure-guided lead optimization.

### 3.4. Identification of New PAMs

In LBVS, 44 compounds have PLANTS scores better than −75, 18 of them are lower than −100 and 11 display an IFP similarity better than 0.7, suggesting their interactions with GLP-1R resembled NNC0640 and PF-06372222. Experimental validation showed that compounds C10 (EN_Z1424437838), C11 (EN_Z1445206940), C13 (EN_Z18696867), C22 (EN_Z26483797), C23 (EN_Z28052152), C26 (EN_Z317215770) and C31 (VM_STL480883) positively modulated GLP-1 potency (Figure 5, Table 2 and Table 3). Figure 6A indicates that C22, C26, C31, C39, C13 and C11 enhanced ligand binding, with C22 being the most potent (IC_50_ = 0.63 μM). They also facilitated GLP-1-induced intracellular cAMP accumulation to various extents (Figure 6C), e.g., 20 μM C22 increased the GLP-1 potency from 34 pM (EC_50_) to 9.8 pM.

C22 and C26 were initially predicted to locate at an extra-helical binding site similar to that of NNC0640 and PF-06372222. With excellent PLANTS and IFP scores, they were able to penetrate into the TM6-TM7 cleft using one arm with the formation of at least two hydrogen bonds (S^6x41b^ and N^8x47b^), while pointing to the TM5-TM6 interface with massive hydrophobic interactions using another arm. In addition, there is one additional hydrogen between C26 and T355^6x44b^. It is interesting that the intracellular region of TM6 appears to regulate the functionality of interacting ligands: NAMs including NNC0640 and PF-06372222 restrict the outward movement of the TM6, a key feature of receptor activation, whereas compound 2 and BETP covalently bind to C347^6x36b^ and act as ago-allosteric modulators (ago-PAMs) [7,22,53,54]. To verify this hypothesis, we mutated two key residues at TM6, C347^6x36b^ and T355^6x44b^, and found that the effects are variable and ligand-dependent. The compounds tested retained their binding affinities to the C347F mutant (Figure 6C), which is different to the abolishment shown by compound 2 [7], indicating that they are not covalently bound to C347^6x36b^, consistent with our observations made in the cAMP accumulation assay (Figure 6D).

To explore the SAR of the nine allosteric modulators discovered in this study (two NAMs and seven PAMs), we performed a structural similarity search in the Chinese National Compound Library database and identified 54 compounds with a high 2D Tanimoto similarity (>0.85) to the nine discovered compounds. These compounds were subjected to both cAMP accumulation and whole-cell binding assays (Table 4). CD3532-B002 and CD3559-D005, two analogues of C11 without one negatively charged atom, displayed a similar enhancement of ligand binding as well as GLP-1-induced intracellular cAMP accumulation. JK1719-D011 weakened the ligand binding, which is different from that observed with its analogue C13, especially considering their structural difference is only limited to the length of the linker. A similar phenomenon was also seen with CD1652-A009 and its analogue C26. These results indicate that the subtle difference in the chemical structures of allosteric modulators may impact their pharmacological profiles.

## 4. Conclusions

The current study was conducted with an aim of developing computational screening protocols to discover new allosteric modulators of GLP-1R in the lipid environment. By adopting MD simulation-derived lipids, we successfully built a membrane environment rich of lipids for molecular docking and SBVS. Through generating a ligand-receptor interaction pattern, LBVS identified seven PAMs.

## Figures and Tables

**Figure 1 biomolecules-11-00929-f001:**
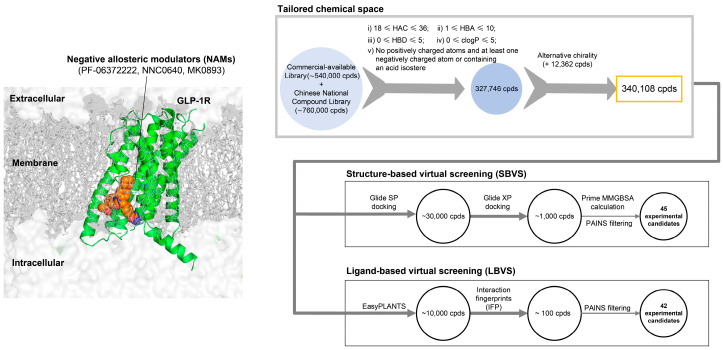
Schematic representation of the GLP-1R allosteric modulator discovery process with structure-based virtual screening (SBVS) and ligand-based virtual screening (LBVS). By analyzing the reported allosteric modulators (PF-0637222, NNC0640 and MK0893), a tailored library consisting of 340,108 compounds underwent the following two different protocols: structure- and ligand-based virtual screening.

**Figure 2 biomolecules-11-00929-f002:**
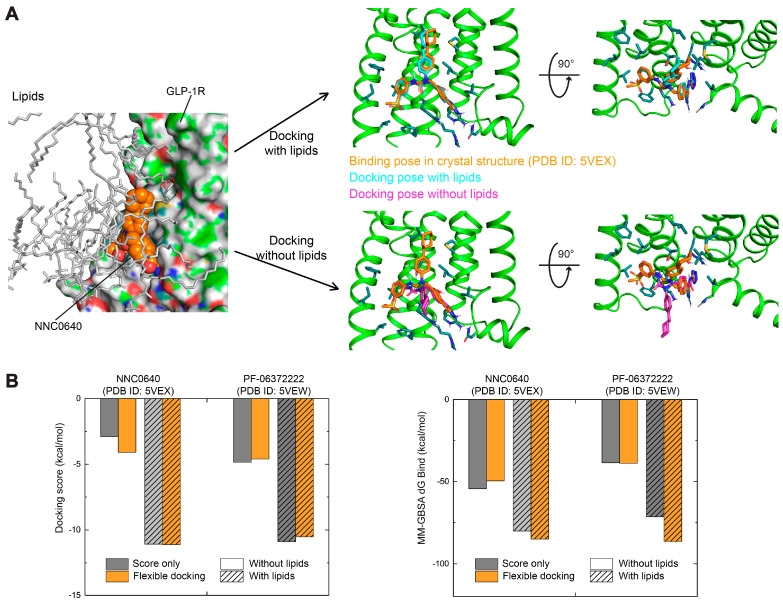
Evaluation of molecular docking protocols on re-docked negative allosteric modulators NNC0640 and PF-06372222 to the allosteric extra-helical binding site of GLP-1R [7]. (**A**) Comparison of docking poses of NNC0640 in GLP-1R by docking protocol with or without the consideration of lipids. The crystal structures of human GLP-1R TMD in complex with PF-06372222 (PDB code: 5VEW) [7] and NNC0640 (PDB code: 5VEX) [7] were obtained from the Protein Data Bank [36]. Carbon atoms of NNC0640 in crystal structure [7], traditional Glide XP docking and Glide XP docking within lipids are colored orange, cyan and magenta, respectively. (**B**) Comparison of docking scores and calculated binding free energies (MM-GBSA dG Bind) of NNC0640 and PF-06372222 to GLP-1R with or without considering the lipid contribution. Molecular docking and binding free energy calculation were performed using Glide and Prime MM-GBSA modules in Schrödinger Suite 2017-3, respectively. Docking score and MM-GBSA dG Bind are in kcal/mol.

**Figure 3 biomolecules-11-00929-f003:**
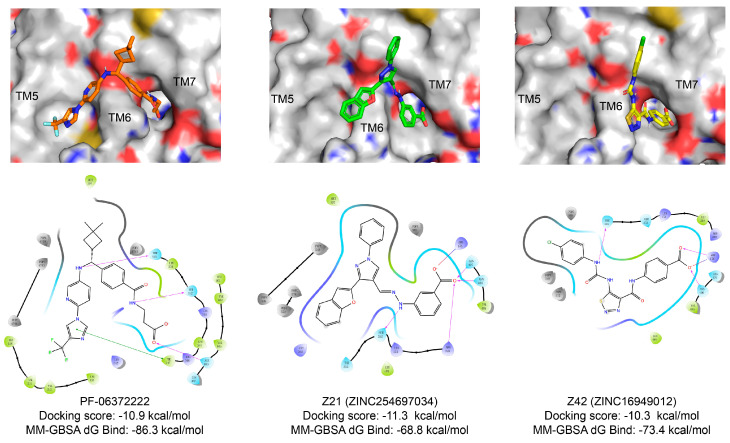
Comparison of binding modes of representative hits from SBVS and PF-06372222. Carbon atoms of PF-06372222, Z21 and Z42 are colored orange, green and yellow, respectively. Docking score and MM-GBSA dG Bind are in kcal/mol.

**Figure 4 biomolecules-11-00929-f004:**
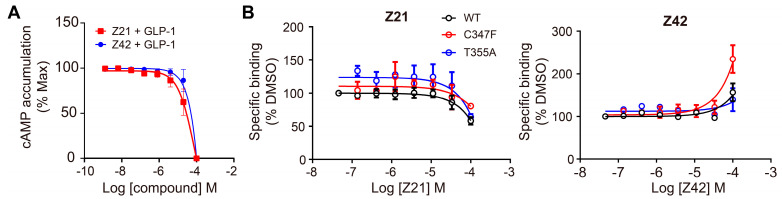
Experimental validation of selected compounds identified using SBVS. (**A**) Dose-dependent inhibition curves of Z21 and Z42 on cAMP activity induced by constant concentration of GLP-1 (0.08 nM for Z21, 0.05 nM for Z42) in GLP-1R expressing CHO-K1 cells. (**B**) Binding of Z21 or Z42 to GLP-1R or its mutants in competition with radiolabeled GLP-1. Data are presented as means ± S.E.M. of three independent experiments.

**Figure 5 biomolecules-11-00929-f005:**
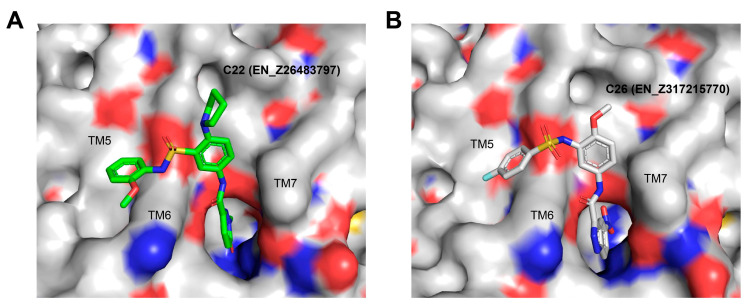
Proposed binding modes of representative hits from LBVS. (**A**,**B**) Carbon atoms of C22 and C26 are colored green and gray, respectively.

**Figure 6 biomolecules-11-00929-f006:**
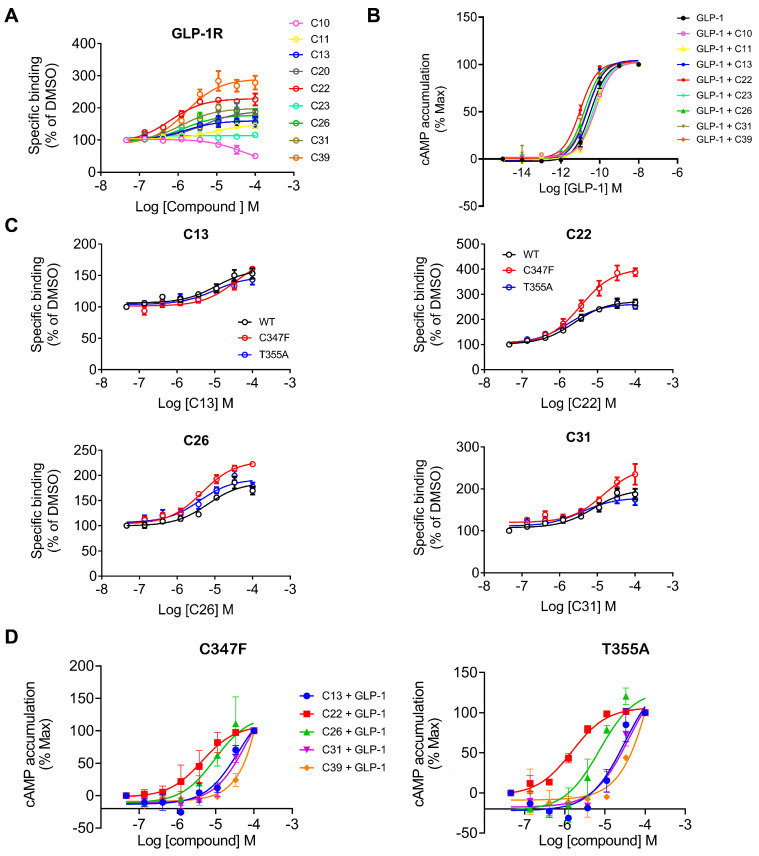
Experimental validation of selected compounds identified using LBVS on GLP-1R. (**A**) Binding of hit compounds to GLP-1R or its mutants in competition with radiolabeled GLP-1. (**B**) cAMP activity induced by different GLP-1 concentrations in the presence of hit compounds (20 μM) in GLP-1R expressing CHO-K1 cells. (**C**) Binding of four hit compounds to GLP-1R mutants (C347F and T355A) in competition with radiolabeled GLP-1. (**D**) Effects of binding-pocket mutations (C347F and T355A) on the allosteric modulation of selected compounds in cAMP accumulation elicited by 0.02 nM GLP-1. Data are presented as means ± S.E.M. of three independent experiments.

**Table 1 biomolecules-11-00929-t001:** Experimentally validated new NAMs based on SBVS approach ^a^.

ID	Name	Chemical Structure	cAMP Accumulation		Binding	
			pEC_50_ ± SEM	E_max_ (% WT ^b^)	pIC_50_ ± SEM	Span ± SEM
Z21	Z49584845	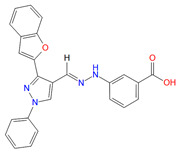	4.12 ± 0.38	97.02 ± 0.81	4.72 ± 0.24	75.3 ± 12.57
Z42	STL446272	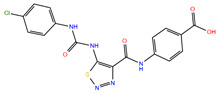	N.A. ^c^	99.72 ± 0.80	N.B. ^d^	N.B. ^d^

^a^ All data were fitted with a three-parameter logistic curve to obtain pEC_50_ and pIC_50_ values. Data represent means ± S.E.M. of at least three independent experiments performed in duplicate.; ^b^ WT, wild-type; ^c^ N.A., not active.; ^d^ N.B., no binding.

**Table 2 biomolecules-11-00929-t002:** Chemical structures and receptor interaction fingerprints of the compounds identified by LBVS approach.

ID	Name	Chemical Structure	PLANTS Score	IFP Score	S^6x41b^	N^8x47b^	T^6x44b^	L^6x48b^	L^6x43b^
C22	EN_Z26483797	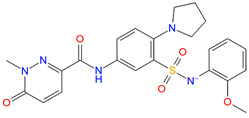	−104.19	0.61	1	1	0	0	1
C26	EN_Z317215770	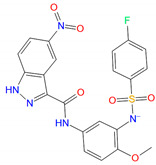	−90.28	0.69	1	1	1	0	1
C10	EN_Z1424437838	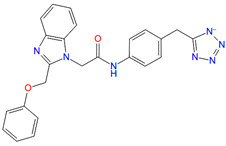	−105.18	0.75	1	1	1	0	1
C11	EN_Z1445206940	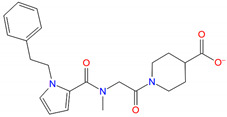	−107.69	0.72	1	1	0	0	1
C13	EN_Z18696867	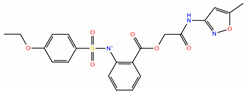	−92.03	0.61	1	1	1	0	0
C23	EN_Z28052152	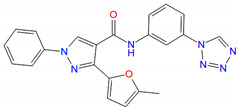	−75.45	0.63	1	1	1	1	0
C31	VM_STL480883	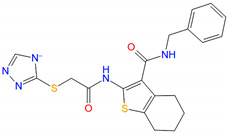	−95.23	0.63	1	1	1	0	1

**Table 3 biomolecules-11-00929-t003:** Experimentally validated new PAMs based on LBVS approach ^a^.

ID	cAMP Accumulation		Binding	
	pEC_50_ ± SEM	E_max_ (% WT ^b^)	pIC_50_ ± SEM	Span ± SEM
C22	6.3 ± 0.29	96.9 ± 4.56	6.2 ± 0.11	−135.7 ± 8.4
C26	5.4 ± 0.31	89.4 ± 3.72	5.85 ± 0.32	−82.09 ± 13.44
C10	3.3 ± 0.47	92.93 ± 4.4	4.34 ± 0.23	76.17 ± 16.21
C11	N.A. ^c^	N.A. ^c^	5.05 ± 0.22	−46.37 ± 5.83
C13	4.8 ± 0.35	76.6 ± 6.24	5.83 ± 0.24	61.32 ± 7.59
C23	4.8 ± 0.21	98.5 ± 3.6	N.B. ^d^	N.B. ^d^
C31	5.0 ± 0.81	91.0 ± 0.95	5.89 ± 0.13	−102.5 ± 6.85

^a^ All data were fitted with a three-parameter logistic curve to obtain pEC_50_ and pIC_50_ values. Data represent means ± S.E.M. of at least three independent experiments performed duplicate; ^b^ WT, wild-type; ^c^ N.A., not active.; ^d^ N.B., no binding.

**Table 4 biomolecules-11-00929-t004:** Experimentally validated analogues of new PAMs ^a^.

Name	Chemical Structure	cAMP Accumulation		Binding	
		pEC_50_ ± SEM	E_max_ (% WT ^b^)	pIC_50_ ± SEM	Span ± SEM
CD3532-B002	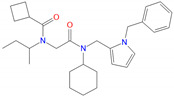	4.8 ± 0.01	92.2 ± 3.9	N.B. ^c^	N.B. ^c^
CD3559-D005	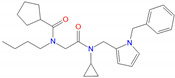	5.1 ± 0.07	78.2 ± 13.6	5.62 ± 0.63	35.25 ± 11.50
JK1719-D011	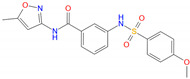	4.2 ± 0.28	126.1 ± 4.2	4.91 ± 0.37	59.69 ± 15.13
CD1652-A009	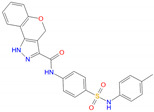	4.8 ± 5.3	114.9 ± 8.6	5.39 ± 0.33	51.92 ± 9.60

^a^ All data were fitted with a three-parameter logistic curve to obtain pEC_50_ and pIC_50_ values; ^b^ WT, wild-type; ^c^ N.B., no binding.

## Data Availability

Data are contained within the article or Appendix A.

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
