# Peer review of "Discovery of Novel Allosteric Modulators Targeting an Extra-Helical Binding Site of GLP-1R Using Structure- and Ligand-Based Virtual Screening"

_biomolecules, 2021, doi:10.3390/biom11070929_

Round 1

Reviewer 1 Report

The manuscript by Zhou et al. describes the discovery of novel allosteric modulators of GLP-1R using structure- and ligand-based virtual screening. Two negative allosteric modulators and seven positive allosteric modulators were identified and verified by radiolabeled ligand binding and cAMP accumulation experiments. The methods are suitable for obtaining the meaningful results and conclusions. Overall, the manuscript is well written and the results are clearly presented.

I have a few concerns, which should be carefully considered and addressed before the manuscript can be recommended for publication:

  1. The description of the ligand-based virtual screening is a little confusing. For example, why and how the cutoff values for PLANTS and IFP score are chosen are not discussed. It is also unclear how the final 42 compounds are obtained from the top 100 compounds after clustering and visual inspection.
  2. Why were two negative allosteric modulators and seven positive allosteric modulators identified from structure-based and ligand-based virtual screening, respectively? Given that two NAMs PF-06372222 and NNC0640 were used in the ligand-based virtual screening, one would expect new NAMs rather than PAMs to be discovered from LBVS. On the contrary, structure-based screening should have identified ligands with more diverse functionalities.
  3. No units are provided for any docking scores or binding free energies, which might indicate more serious flaws in conducting the study. Were all the calculations and simulations conducted correctly, and can we trust all the reported data?

Other minor points:

P1, line 30, discovery --> discover

P4, line 181, atexcitation --> at excitation

P9, line 320, as ago allosteric --> as an allosteric

P10, line 339, six or seven PAMs?

Reviewer 2 Report

The authors present a report on the identification of allosteric modulators for the extrahelical binding site of the glucagon-like peptide-1 receptor. Both structure- and ligand-based methods were employed, and active ligands were found using both methods. A binding assay to the receptor was used as well as a cell-based assay monitoring cAMP, a secondary messenger.

The manuscript is well written and in a logical manner. There are some typos/mistakes that need to be cleaned up.

There are three main issues that need to be addressed:

  1. The molecular structures of the experimentally verified compounds need to be shown
  2. The experimental data needs to be tabulated for clarity

3. There is no Structure Activity Relationship analysis

4. The hits are not particularly potent, and no analogues of the most potent hits tested. I recommend that the authors purchase structural analogues of the hits and test them for SAR.

I recommend a major revision of this manuscript.

Small things:

  1. No reference is given for crystal structure
  2. No references are given for the Protein Data Bank

Reviewer 3 Report

The paper by Zhou et al. is a very nice example of rational drug design. The manuscript is well written and the reader can follow step by step how Authors identified six promising GLP-1 positive allosteric modulators. Materials and methods are exhaustively described, images are of good quality and results are sounding. Above all, I appreciated the computational hypothesis has been validated experimentally. This point is very important since it demonstrates that the study it is not a simple computational speculation and the strategy followed produced objective results.

In my opinion the manuscript should be published as it is.

Author Response

We appreciate the reviewer’s positive comments.

Round 2

Reviewer 2 Report

The authors have improved their manuscript based on the reviewers' comments.

I recommend this manuscript for publication.